# Multiomic Data Integration in the Analysis of Drought-Responsive Mechanisms in *Quercus ilex* Seedlings

**DOI:** 10.3390/plants11223067

**Published:** 2022-11-12

**Authors:** Víctor M. Guerrero-Sánchez, Cristina López-Hidalgo, María-Dolores Rey, María Ángeles Castillejo, Jesús V. Jorrín-Novo, Mónica Escandón

**Affiliations:** Agroforestry and Plant Biochemistry, Proteomics and Systems Biology, Department of Biochemistry and Molecular Biology, University of Cordoba, UCO-CeiA3, 14014 Cordoba, Spain

**Keywords:** *Quercus ilex*, multiomics, integrative approach, drought, proteomics, metabolomics, transcriptomics, DIABLO, PCA

## Abstract

The integrated analysis of different omic layers can provide new knowledge not provided by their individual analysis. This approach is also necessary to validate data and reveal post-transcriptional and post-translational mechanisms of gene expression regulation. In this work, we validated the possibility of applying this approach to non-model species such as *Quercus ilex*. Transcriptomics, proteomics, and metabolomics from *Q. ilex* seedlings subjected to drought-like conditions under the typical summer conditions in southern Spain were integrated using a non-targeted approach. Two integrative approaches, PCA and DIABLO, were used and compared. Both approaches seek to reduce dimensionality, preserving the maximum information. DIABLO also allows one to infer interconnections between the different omic layers. For easy visualization and analysis, these interconnections were analyzed using functional and statistical networks. We were able to validate results obtained by analyzing the omic layers separately. We identified the importance of protein homeostasis with numerous protease and chaperones in the networks. We also discovered new key processes, such as transcriptional control, and identified the key function of transcription factors, such as *DREB2A*, *WRKY65*, and *CONSTANS*, in the early response to drought.

## 1. Introduction

Holm oak (*Quercus ilex* L.) is the dominant tree species in the Mediterranean forest and the key element of the Spanish agrosilvopastoral “*dehesa*”, with both ecological and economic importance [1]. Currently, several anthropogenic and environmental factors, such as aging trees, overexploitation, and abiotic and biotic stresses (drought, *Phytophthora cinnamomi*, etc.), threaten the sustainability and survival of *Q. ilex* and related ecosystems [2,3]. A combination of these stresses triggers the so-called Holm oak decline syndrome, leading to an increase in tree mortality and reducing the forested area. Holm oak is quite tolerant to drought and high temperatures [4]. However, the current climate change scenario, characterized by a drastic decrease in rainfall and a severe rise in temperatures, means that many areas where this species is found would no longer be suitable for its survival [5,6]. In addition, seedlings are less tolerant than adult trees, making water deprivation the main cause of the failure of reforestation programs [4]. For this reason, identifying and characterizing phenotypes more resilient to drought is a priority in plant breeding, management, and conservation programs in the Mediterranean area.

Since holm oak is a non-domesticated species and considering its biological characteristics (long-lived cycle, allogamy, and pollen dispersal by wind), the only viable alternative for its breeding is to exploit the biodiversity to select elite resilient genotypes [7,8]. For this purpose, it is necessary to characterize its inter- and intrapopulation variability and study its molecular mechanisms of resisting and tolerating biotic and abiotic stresses [9,10]. Although holm oak is a non-model species, the first draft of the genome has been recently published [11]. Furthermore, in recent years, great advances have been made through omic techniques [10,12,13], for example, a reference transcriptome with a fairly complete collection of holm oak genes [13] and a protein database that allows proteomic analysis through mass spectrometry technologies [14] have been obtained. In addition, the acorn and leaf metabolomes have been recently described [8,15].

When studying biological systems, traditionally, the different molecular levels analyzed by each omic technology have been studied by applying a reductionist and independent vision [16,17]. However, although each omic technology offers a lot of information, individual analysis cannot describe the entire biological complexity resulting from the relationships between the different molecular layers [18]. In addition, the molecular levels are influenced by the environment [19]. For this reason, the current trend is to combine different omic technologies to get a holistic view of the studied system [16,20]. This new approach provides a better understanding of the biology of the species and allows one to select stress-related resilience genes or gene products as markers in breeding programs that can help in the conservation of an endangered species, such as the holm oak.

The development of bioinformatics software has allowed us to carry out a joint analysis of the large amount of data generated by omic technologies to establish statistically significant relationships between the different molecular layers [21]. However, interpreting the results of such analysis remains a challenge even in model species. Some molecular analyses have been performed on holm oak [12] using a non-targeted omic analysis [8,22,23,24] or a non-targeted multiomic approach [14,25]. Our group has previously analyzed the effect of drought on *Q. ilex* and its response to drought by using morphology, classic biochemistry, physiology, and omic approaches [7,8,14,22,26,27,28,29]. Despite numerous studies, a global non-targeted multiomic analysis of the drought response of *Q. ilex* species has not been performed to date.

In this work, using a systems biology approach, we have evaluated the effect of drought stress on six-month-old seedlings to characterize the molecular mechanisms triggered under the stress response (Figure 1). With that aim, we performed a non-targeted integrative multiomic analysis (transcriptomics, proteomics, and metabolomics) of the omic data generated by our group in previous studies [8,14,22], carrying out complementary interaction network analysis. In addition, a multivariate principal component analysis (PCA) was performed for comparison with the non-targeted integrative multiomic analysis (transcriptome, proteome, and metabolome) and to supplement it. Non-targeted integrative multiomic analysis allowed us to evaluate and categorize the transcripts, proteins, and metabolites that define the response to drought in *Q. ilex* seedlings and to obtain potential biomarkers of resilience to be validated and employed in selecting elite, resilient genotypes for breeding and reforestation/conservation programs.

## 2. Results and Discussion

### 2.1. Comparison of Principal Component Analysis (PCA) and DIABLO Multiomic Data Analysis

To assess the response of holm oak to drought from a systems biology perspective and to see how a non-targeted multiomic approach can provide new knowledge on the biology of this species, we carried out different multivariate analyses. The first dimensionality reduction approach was based on the popular principal component analysis (PCA) using the entire dataset (47,569 transcripts; 3312 proteins; 342 metabolites) (Figure 2A, Appendix A), and the other PCA was based on only those with differences between treatments with a *p*-value lower than 0.05 (4574 transcripts; 562 proteins; 81 metabolites) (Figure 2B, Appendix A) in the three datasets. The number of variables was reduced to less than 6000 using the significant variables, reducing the noise level. However, the initial proportion of each omic layer was still retained. In the PCA plots (Figure 2A,B), well-watered seedlings (control) and seedlings exposed to drought-like conditions for two lengths of time (D1, 17 days and D2, 24 days) were separated when the original dataset was filtered for differential variables with *p*-value < 0.05 (Figure 2B). In both analyses, transcripts were the variables with more weight in components 1 and 2, whereas proteins and metabolites were represented within these variables with more weight in component 2 (Appendix A). 

In the second dimensionality reduction approach, we performed a DIABLO (Data Integration Analysis for Biomarker discovery using Latent variable approaches for Omics studies) analysis, a method of integration and interrelation of multiple omic layers (Figure 2C and the partial DIABLO plots of each omic dataset). In DIABLO, the variables with the highest sample discrimination power are selected and the optimum number of variables within each omic dataset is chosen, thus reducing the complexity of the thousands of variables in the omic data. In our data, the optimal number of variables within the three analyzed omic layers for each component were: 350 transcripts (250 in component 1 (comp1) and 100 in component 2 (comp2)) out of a total of 47,569; 514 proteins (300 in comp1 and 300 in comp2, with 86 in both components) out of a total of 3312; and 71 metabolites (62 in comp1 and 10 in comp2, with 1 in both components) out of a total of 342 (Appendix A). Comparing the variables selected by DIABLO with the variables with the highest loadings in PCA, we could observe that the three omic datasets were more equally represented in DIABLO, which did not show a bias toward the largest dataset, i.e., transcriptomics.

DIABLO analysis also succeeded in separating the seedlings exposed to drought-like conditions for the two lengths of time (D1 and D2); Figure 2C. Comp1 separated D2 from the rest, and comp2 separated D1. Controls, on the other hand, were similar, unlike in PCA. Furthermore, in DIABLO analysis (Figure 2C), we can see how much variance was explained by each omic layer separately, seeing that the variables selected can do a relatively good job of separating the samples in the individual omic layers (Figure 2C, partial DIABLO plots). Metabolites were the major contributors in comp1 (39.41%), whereas transcripts were the major contributors in comp2 (38.68%). Proteins presented similar percentages (around 30–35%) in both components. Finally, the integration of the three omic layers allowed a better grouping between replicates, as seen in the boxes in Figure 2C. 

To identify the dominant trends in the variables selected by DIABLO for treatment separation, a heatmap was created (Figure 3; Appendix A). We can observe that the magnitude of the molecular changes due to the response to drought stress clustered treatments and replicates. Five clear clusters can be observed (Figure 3, from top to bottom): (i) early transient overexpression/accumulation in seedlings exposed to drought-like conditions (D1, early response); (ii) overexpression/accumulation in controls (C1 and C2); (iii) transient overexpression/accumulation at a later time in seedlings exposed to drought-like conditions (D2, later response); (iv) overexpression/accumulation at time 2 in both control seedlings and seedlings exposed to drought-like conditions (C2 and D2); and (v) permanent changes with overexpression/accumulation at both times in seedlings exposed to drought-like conditions (D1 and D2, permanent stress changes). Among these clusters, drought clusters are the most interesting and form the most abundant groups (D1, D1 and D2, and D2, from the highest to the lowest). In these main clusters, we can see a high percentage of transcripts and proteins related to cellular homeostasis, such as proteases and chaperones, necessary to maintain cellular protein function during drought stress [30]. D1 and D2 (permanent stress changes) is the group with the highest number of variables included in cellular homeostasis (small heat shock proteins, ClpBs, heat shock 70/80/90 Kda proteins, molecular chaperone GroEl, LEA, etc.), followed by the D2 (later response) group (ClpB2, small heat shock proteins, heat shock 70 Kda proteins, and DNAJ heat shock family proteins) and to a lesser extent the D1 (early response) group (heat shock 70/83 Kda proteins). In addition, redox homeostasis shows a higher percentage of variables in the D1 and D2 (permanent stress changes) group (thioredoxin H2/H9/F-type or reductase 1, nucleoredoxins, and several glutathione S-transferases or L-ascorbate peroxidases). Protein and redox homeostasis are key to the survival of plants under stress [31,32,33]. Heat shock proteins (HSPs) and proteases play a fundamental role in protecting plants under stress since they are responsible for maintaining protein conformation and folding or degrading damaged proteins [32,34]. In drought, in particular, the importance of these transcripts and proteins had already been seen in previous studies [14]. Moreover, antioxidant machinery activation is key to counteracting the overaccumulation of reactive oxygen species during stress [33,35].

At the metabolomic level, we have observed a slight metabolite accumulation in the later response (D2) group, where more metabolites were included, showing a higher abundance of several amino acids and peptides (Figure 3; Appendix A), highlighting L-proline and L-phenylalanine. L-proline accumulation is a common response to several abiotic stresses, functioning as an osmotic agent and radical scavenger [36]. L-phenylalanine accumulation is related to drought response, and its accumulation has already been observed in other species of the genus *Quercus*, such as *Q. suber* [37]. This compound is an essential precursor of numerous secondary metabolites (flavonoids, coumarins, monolignols, etc.) as the initiator of the core phenylpropanoid biosynthesis pathway. This may be due to the importance of phenolic compounds in the biology of this species, being the most represented compounds in the *Q. ilex* metabolome [8].

### 2.2. DIABLO and STITCH Interaction Networks and Their Biological Interpretation

To visualize the interactions between the different omic layers, two types of interaction networks were created: a statistical one, obtained by DIABLO (Figure 4), and a functional one, using STITCH (chemical–protein interaction network) (Figure 5) [38]. Both networks were built with the variables selected by comp1 and comp2 of DIABLO (350 transcripts, 514 proteins, and 71 metabolites), finding a high percentage of homologs in the STRING/STITCH databases (309 transcripts, 501 proteins, and 71 metabolites). The networks presented were reduced to the strongest interactions (DIABLO) or the highest confidence (STITCH), established at 0.9, in both cases. At these thresholds, the variables selected in the two networks were different, providing two complementary views of the *Q. ilex* drought response. The DIABLO network contains mainly early response variables (Figure 4B), whereas STITCH contains mainly permanent changes during drought (D1 and D2) (Figure 5B). 

Next, we discuss the key gene products and interconnections obtained from the integrated analysis and potentially of relevance in the response to drought. 

As shown in Figure 4B and Figure 5B, in both networks, protein biosynthesis and homeostasis are the main processes, forming clear sub-networks made up of a large number of variables. STITCH shows a clear main ribosomal network, another one including components of the proteasome complex, and a third one including chaperones (Figure 5B). In DIABLO, there is no clear distribution except for the small network on the right, associated with stress and cellular homeostasis and including chaperones and proteases (Figure 4B). Stress response and protein homeostasis processes show variables in both networks, with several candidates in both interaction networks (such as the 17.5 kDa HSP family protein, several HSP70, and proteasome subunits). This group of proteins accumulated in D1, D1 and D2, and D2 (Figure 4B) has also been described in previous work [14], with a greater number of proteins of this family identified in this work, giving a more complete vision of the HSPs implicated in drought stress. In the small DIABLO network on the right, HSPs along with other stress proteins are interconnected by some metabolites, such as emodin and amino acids (L-proline and homoarginine). The anthraquinone emodin is a secondary compound affected by abiotic environmental factors, such as osmotic stresses (drought and salinity) and light intensity, [39] and its external application triggers a higher expression of HSPs [40]. L-proline (accumulated in D2; Figure 4B and Figure 5B) acts not only as an osmoticum (a substance that acts to supplement osmotic pressure in a plant) [36] but also as a molecular chaperone, preserving protein integrity and speeding up the activity of many enzymes [41], which could explain the direct relationship shown in the DIABLO network between these metabolites and HSPs. Furthermore, proline residues at the N-terminus of alpha helices have been shown to contribute to protein thermostability [42]. 

With respect to protease activity, we found two casein lytic proteinases, ClpB2 (qilexprot_8964), common in the two networks, and *ClpB3* (contig-10000000), in the DIABLO network. An ATP-dependent Clp protease (qilexprot_22489) was also observed in the DIABLO network. These chaperones were highly represented in response to drought, and some of them were selected as candidates for drought tolerance in a study by Guerrero-Sanchez et al. [14]. Furthermore, these chaperones, together with other types of proteins, such as FTSH6 (qilexprot_61641, present in the small DIABLO network, and the other candidate proposed by Guerrero-Sanchez et al. [14]), have been described to be involved in the degradation of the light-harvesting complex of photosystem II (LHC II) during senescence or high light acclimation [43,44]. Loss of water could result in the plant being unable to maintain an adequate temperature and resist the high irradiance, because of which, it is forced to activate defense mechanisms against this type of stress resulting from drought stress in these environmental conditions. 

Regarding the protein biosynthesis process, ribosomal proteins can be found as candidates in both networks (as protein and transcript variables) but are more represented in the STITCH network (the main circular network in Figure 5). Ribosomal proteins, such as 60S ribosomal protein L14-2 (qilexprot_78276), are implicated in several regulatory processes, such as drought [45]. The DIABLO network (Figure 4) showed two transcripts belonging to this group, *40S RIBOSOMAL PROTEIN S13*, and *60S RIBOSOMAL PROTEIN L34*, upregulated in D1. Both transcripts shared the same positive interconnections (as with elongation factors related to the biosynthesis process) and a negative one with 3-coumaric acid (precursor of the phenylpropanoid pathway or the phenylpropanoid–acetate pathway).

In addition to the predominant processes mentioned above, which to a greater or lesser extent have been seen in the previous analysis of the different omic layers [8,14,22], we were able to observe important new pathways or processes related to the response to drought, which will be discussed next.

When a plant faces unfavorable conditions, for example, drought, to survive it must modulate gene expression to optimize the response to overcome such conditions. This task is performed by transcription factors (TFs), which trigger the conversion of stress-induced signals into cellular responses [46]. A novelty in the DIABLO network, we can see a high number of TFs upregulated in early response (D1, Figure 4B), some related to stress (*DREB2A*-interacting protein 2 isoform *1,* probable *WRKY* transcription factor 65, and heat stress transcription factor A2-b) [47], some related to the circadian clock (*CONSTANS*-like zinc finger protein, zinc finger protein *ZPR1*, and *Zf-ZPR1* domain-containing protein) [48,49], and some related to growth and development processes (transcription factor *bHLH74* and zinc finger *BED* domain-containing protein *DAYSLEEPER*) [50,51].

*DREB2A* (contig-1429000013) is a transcription factor that interacts with a cis-acting dehydration-responsive element (DRE) sequence and activates the expression of downstream genes involved in drought– and salt–stress response in *Arabidopsis thaliana* [52]. *DREB2A*, whose tendency changes after 17 days of drought, shows a relatively rapid plant response by triggering stress response mechanisms before the fluorescence drops below 50%. Furthermore, Arabidopsis *DREB2A* has a dual role in combating both water and heat-shock stress responses [52]. Drought and heat stress are highly interrelated. Both stresses are often combined under field conditions, and between them, an overlap of heat- and drought-responsive genes has been observed [53]. Therefore, the presence of heat stress TFs (such as heat stress transcription factor A2-b) is common in drought. Heat stress transcription factors are important transcription factors in both heat and drought stresses [47]. The transcriptional expression level of HSPs is regulated by heat stress TFs, as well as regulating other TFs, such as *DREB2A* [54]. However, little is known about heat stress transcription factor A2-b and its involvement in drought.

Lastly, among stress TFs, *WRKY* is a large plant family of TFs, which is induced by numerous abiotic stresses, such as drought and heat stress. In the DIABLO network, probable *WRKY* transcription factor 65 showed many connections with other elements in the upregulated network in the early response (D1, Figure 4B), positioned at the top of the network. Several *WRKYs* have been identified that enhance tolerance [55,56,57,58]. Recently, Huo et al. [59] found that *ZmWRKY65*, a gene in the maize, enhanced drought tolerance in *Arabidopsis*, proposed as a multifunctional factor that can integrate different stress signals in plants.

Candidates related to flowering and the circadian clock appear to be key in the short-time drought response (D1). Transcript factor *CONSTANS* (contig-4162000011) (Figure 4) promotes flowering and has been found to be related to the control of flowering and the circadian clock [48]. Moreover, in the network, we can see the presence of *GIGANTEA* (contig-291000017) (Figure 4 and Figure 5) linked with adagio protein 3 (contig-440000008) (Figure 5). The complex forming by *GIGANTEA* and adagio protein 3 regulates *CONSTANS* expression [60]. Given the age of the plants, the flowering process is not a viable hypothesis, being more related to the involvement of *CONSTANS* in regulating other genes, such as *P5CS2* related to proline biosynthesis (over-accumulated in later response D2). Furthermore, *GIGANTEA* is a well-known protein with a role in drought tolerance. *GIGANTEA* is a key regulator of the photoperiod and the circadian cycle and is also involved in the signaling pathways for various abiotic stresses, such as the regulation of ABA synthesis [61].

Finally, we can see that an antioxidant response is necessary for the detoxification of the ROS produced during the whole drought stress (D1 and D2). This increase in antioxidant enzymes had been seen in previous studies [8,24], but the candidates responsible for this response were not clearly identified. In the DIABLO network, the presence of relevant enzymes, such as APX2 (qilexprot_28343) and glutathione S-transferase (qilexprot_71801), was observed. Ascorbate peroxidase (APX) isoenzymes play a key role in the ascorbate–glutathione cycle, which is the major hydrogen peroxide detoxifying system [62]. Rossel et al. [63] showed a constitutively higher expression of cytosolic-APX2-enhanced tolerance to drought in an *Arabidopsis* mutant. The other enzyme, glutathione S-transferase, is a protein involved in the metabolism of oxidative stress caused by the ROS produced because of water deprivation and high irradiance [64]. This protein is related to several proteins according to the DIABLO network (Figure 4), where we can see proteins related to secondary metabolism with possible antioxidant capacity, such as terpenes ((+)-neomenthol dehydrogenase and carotenoid cleavage dioxygenase 4) and flavonoids (NAD(P)H-dependent 6′-deoxychalcone synthase).

## 3. Materials and Methods

### 3.1. Plant Material and Experiment Description

The drought experiment was performed with seedlings of the drought-tolerant *Quercus ilex* species [7]. Six-month-old seedlings obtained from healthy open-pollinated acorns from trees located in Almadén de la Plata (Seville, Andalusia, Spain; 37°52′ N, 6°28′ W) were used. These seedlings had a size of about 10–15 cm and had 8 to 10 adult leaves. The experiment was performed under the typical summer conditions in Andalusia, (mean temperature 37 °C, maximum temperature 44 °C, minimum temperature 19 °C, and relative humidity 40%) under severe drought conditions (withholding water for 28 days) [7]. Well-watered plants (C) were kept at 100% substrate moisture throughout the experiment [22]. Leaves were collected when the chlorophyll fluorescence in seedlings subjected to drought-like conditions decreased by 20% (17 days of drought, D1, early response) and 45% (24 days of drought, D2, late response) with respect to the control, well-watered, seedlings [7]. Three asymptomatic seedlings were analyzed at each sampling point (C1,D1 and C2,D2), each seedling being a biological replicate. Adult leaves without stress symptoms were washed, frozen in liquid nitrogen, and stored at −80 until extraction (metabolites, proteins, and RNA). Detailed information about plant material and the experiment can be found elsewhere [22].

### 3.2. Non-Targeted Transcriptomic Analysis

RNA extraction has been described in Ref. [24], and the high-throughput sequencing Illumina Hiseq 4000 platform (Illumina, San Diego, CA, USA) was used to obtain the libraries. Previously, a de novo transcriptome for *Q. ilex* was generated from raw data obtained from Illumina and Ion Torrent [13,24]. Such raw data, together with the datasets for well-watered seedlings and seedlings exposed to drought-like conditions, were used to assemble all clean *Q. ilex* reads into contigs by using RAY v. 2.3.1 [65]. The assembly structure was evaluated with QUAST v. 5.0.0 [66]. This new version of the *Q. ilex* transcriptome was annotated using Sma3s v. 2 [67] and the UniRef90 database (www.uniprot.org, accessed on 29 June 2019). To quantify the expression of transcripts, RSEM [68] was used to map the filtered reads to the reference transcriptome, obtaining the abundance of each transcript. The complete transcript dataset used in the analysis is available in Appendix A. The transcriptome raw data are available in the NCBI’s Gene Expression Omnibus repository (GEO, accession number GSE145009, https://www.ncbi.nlm.nih.gov/geo/query/acc.cgi?acc=GSE145009, accessed on 15 October 2020).

### 3.3. Non-Targeted Proteomic Analysis

Protein extraction, separation, and mass spectrometry analysis have been described in Ref. [22]. Briefly, proteins were extracted by following the trichloroacetic acid (TCA)/acetone–phenol protocol according to Wang et al. [69]. Protein samples were run 1 cm in 12% sodium dodecyl sulfate–polyacrylamide gel electrophoresis (SDS–PAGE), and the resulting single bands were trypsin-digested and subjected to shotgun proteomics analysis (nano-LC-MS-UHPLC Ultimate 3000-Orbitrap Fusion, Thermo Scientific, San Jose, CA, USA) according to Gómez-Gálvez et al. [70]. MS/MS data were processed employing Proteome Discoverer 2.3 (Thermo, San Jose, CA, USA), the proteins were identified using the SEQUEST algorithm with the species-specific *Q. ilex* transcriptome database [13,24], and the data were enriched with the RNA-seq data generated in this experiment, as described above. Proteins were quantified as described in Ref. [71]. The complete proteomic dataset used in the analysis is available in Appendix A. The protein raw mass spectrometry was deposited in the ProteomeXchange Consortium via the PRIDE partner repository [72] with the dataset identifier PXD017493 (http://proteomecentral.proteomexchange.org/cgi/GetDataset?ID=PXD017493, accessed on 15 October 2020).

### 3.4. Non-Targeted Metabolomics Analysis

Metabolite extraction, separation, and mass spectrometry analysis have been described in Ref. [8]. Briefly, metabolites were extracted as described by Valledor et al. [73] using three replicates of each treatment and a quality control mix. Dried extracts were re-dissolved in 1 mL of 50% methanol and 0.1% formic acid. After filtration, 5 μL of the sample was subjected to chromatographic separation with a Dionex Ultimate 3000 RS UHPLC system (Thermo Fisher Scientific, Bremen, Germany) equipped with an Acquity UPLC BEH (bridged ethyl hybrid) C18 column (1.7 μm; 100 × 2.1 mm) (Waters Corporation, Manchester, UK) held at 40 °C. Chromatography parameters and analysis have been described in Ref. [8]. The software Compound discoverer^®^ (v. 3.1) (Thermo Scientific, San Jose, CA, USA) was used to quantify and identify each feature. The area under the curve of the peaks was used for the relative quantification. Identification was performed by comparing with three different databases included in the software (mzCloud, ChemSpider, and KEGG). The complete metabolomic dataset used in the analysis is available in Appendix A. Metabolite raw data were deposited in the NIH Common Fund’s National Metabolomics Data Repository (NMDR) website, the MetabolomicsWorkbench, https://www.metabolomicsworkbench.org (accessed on 13 September 2020), where it has been assigned the project ID PR001361. The data can be accessed directly via the project DOI 10.21228/M8SQ6J.

### 3.5. Data Preprocessing and Statistical Analysis

Previous preprocessed data of transcripts [14], proteins [14,22], and metabolites [8] were used (Appendix A). Scaled and centered values (z-scores) were subjected to multivariate analysis with the integration of omic levels. Principal component analysis (PCA) and DIABLO (Data Integration Analysis for Biomarker discovery using Latent variable approaches for Omics studies) [74] were conducted with pRocessomic v.1.8 (available on http://github.com/Valledor/pRocessomics, accessed on 29 January 2020). Both analyses are a dimensional reduction approach, but DIABLO selects the optimal number of variables to obtain the maximum discriminant power and provides the inter-relationship between the different omic layers. The *autotune()* function was used in DIABLO to select the optimal number of variables to include from the three integrated omic layers (with significant variable features *p* < 0.05). This function involves comparing multiple models by selecting different numbers of variables within each dataset, choosing the number of variables that best predict their own accuracy [75]. So, 350 transcripts (250 in component 1 (comp1) and 100 in component 2 (comp2)), 514 proteins (300 in comp1 and 300 in comp2, with 86 in both components), and 71 metabolites (62 in comp1 and 10 comp2, with 1 in both components) were selected and included in DIABLO analysis. PCA was performed first with all variables in the datasets (47,569 transcripts; 3312 proteins; 342 metabolites) and then with only the significant variables at *p* < 0.05 (4574 transcripts; 567 proteins; 81 metabolites). Heatmap and plots were made using pRocessomic v.1.8. The networks and visualization were generated by pRocessomic and Cytoscape program 3.9 [76], using DIABLO analysis (DIABLO network, Figure 4) and the StringApp plugin (STITCH network, Figure 5), using the model species *A. thaliana* [77]. The networks were simplified to interactions with a threshold of ±0.9 or higher. In the DIABLO network, −0.97 to −0.9 (negative) and 0.9 to 0.97 (positive) represent the strongest correlations between variables, and in the STITCH network, 0.9 to 1 represents the interactions with the highest confidence.

## 4. Conclusions

We have used an integrated multiomics analysis to generate new information beyond that obtained by using transcriptomic, proteomic, and metabolomic approaches independently of each other, contributing to the knowledge of the responses of *Q. ilex* to drought. In the study of the response to drought, DIABLO provided a better model than PCA and allowed us to interrelate the different omic layers. Within the interaction networks, we confirm the importance of previous candidates in the response to drought, such as the FTsH6 protease, involved in the degradation of the light-harvesting complex of photosystem II, and the ClpB proteases, with several members of this family within both networks. These processes appear to be key in the permanent changes to stress. On the other side, early responses are characterized by the appearance of transcription factors (*DEBR2A, WRK65, CONSTANT*, etc.) up-regulating stress-related genes. The appearance of these TFs could be related to the prioritization of the gene response in the early stress stage and to the increase in the osmoprotectant proline observed in later stages, among others. At the spatiotemporal level, we have seen that some responsive gene products or metabolites are transiently accumulated in early stages (*DEBR2A, WRK65,* and *CONSTANT*) and later stages (such as L-proline), whereas others (such as FTSH6, APX2, and glutathione S-transferase) are permanently accumulated in both sampling times. The latter ones can be proposed as markers of resilience; one of them has been proposed in previous studies.

We have gone one step ahead in the molecular study of the responses of an orphan tree species, *Q. ilex*, to drought. We have shown that it is possible to take the systems biology approach not only in model plants, such as *A. thaliana*, but also in non-model, orphan, and recalcitrant organisms. However, the systems biology approach with non-domesticated and poorly characterized species is challenging and with limitations since the results depend, largely, on the homology of species-specific genes to those of model species, such as *Arabidopsis*.

## Figures and Tables

**Figure 1 plants-11-03067-f001:**
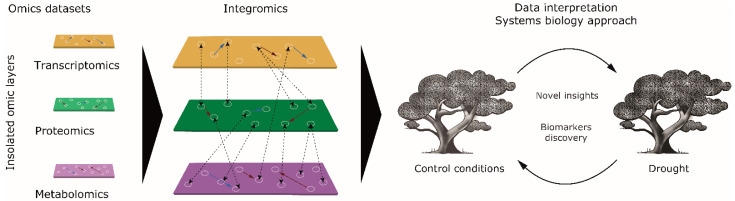
Systems biology data interpretation approach in holm oak. Integration with three omic layers for novel insights and the discovery of biomarkers. The red arrows indicate positive correlations, the blue arrows indicate negative correlations, and the dashed lines indicate interrelations between omic layers.

**Figure 2 plants-11-03067-f002:**
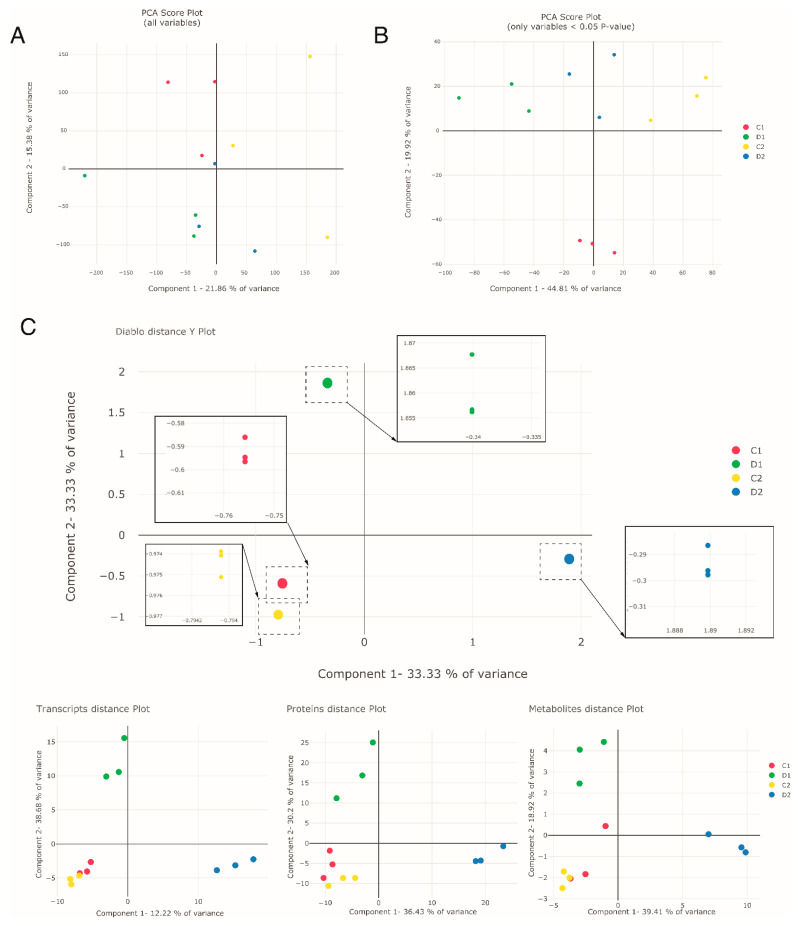
Multivariate PCA and DIABLO analysis of the proteomic, transcriptomic, and metabolomic datasets. (**A**) PCA plot of all variables, (**B**) PCA plot of significant variables, and (**C**) DIABLO distance Y plot and the partial DIABLO plots of each omic dataset of the selected variables (transcripts, proteins, and metabolites). D1 corresponds to exposure to drought-like conditions for 17 days (early response, when chlorophyll fluorescence decreases by 20%), and D2 corresponds to exposure to drought-like conditions for 24 days (later response, when chlorophyll fluorescence decreases by 45%). C1 and C2 are the controls taken at both times.

**Figure 3 plants-11-03067-f003:**
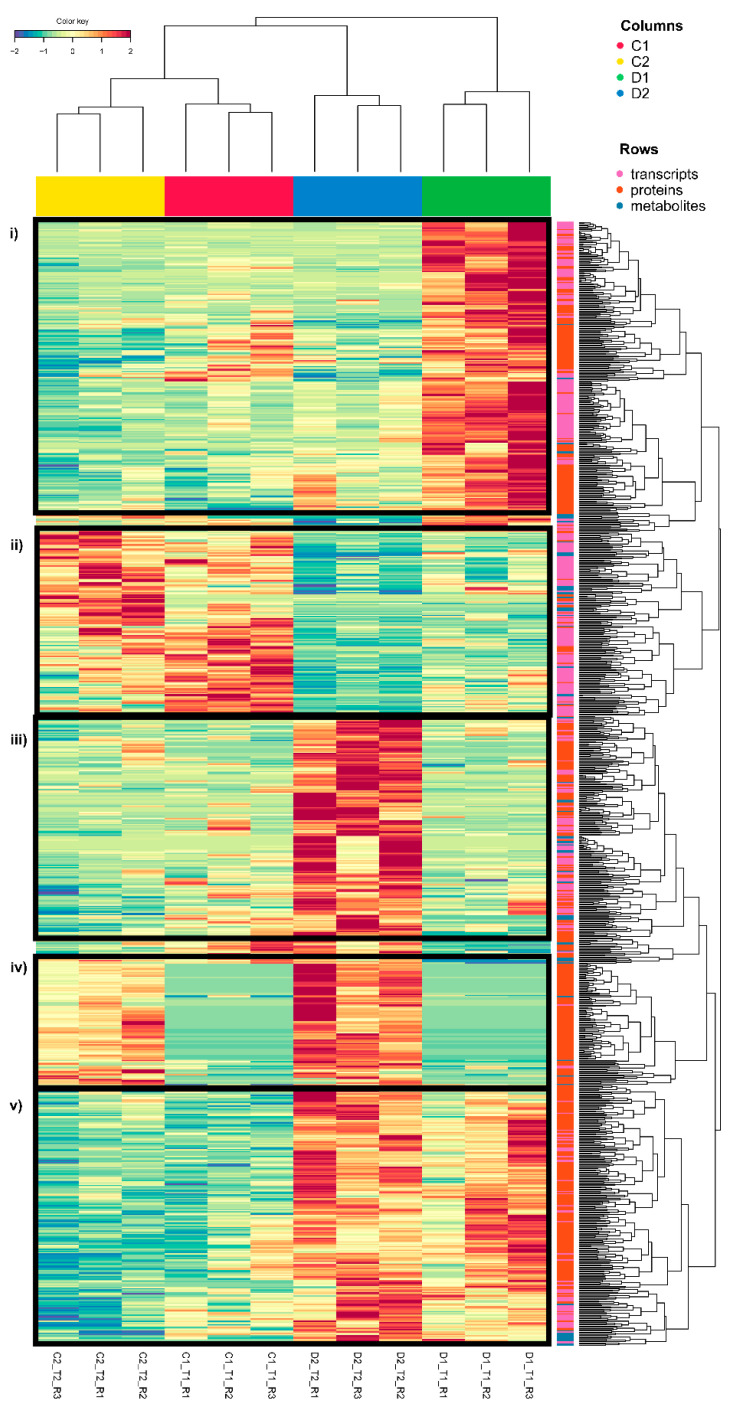
Heatmap of the variables selected for DIABLO: 350 transcripts (250 in comp1 and 100 in comp2), 514 proteins (300 in comp1 and 300 in comp2, with 86 in both components), and 71 metabolites (62 in comp1 and 10 in comp2, with 1 in both components). The five clear clusters have been framed in black (**i**–**v**). The variables selected in the three omic layers are represented in pink (transcripts), orange (proteins), and blue (metabolites) in rows. D1 corresponds to 17 days of drought (early response, when chlorophyll fluorescence decreases by 20%) and D2 corresponds to 24 days of drought (later response, when chlorophyll fluorescence decreases by 45%). C1 and C2 are the controls taken at both times. The figure has been compressed. Appendix A corresponds to the same figure uncompressed with all variable names available.

**Figure 4 plants-11-03067-f004:**
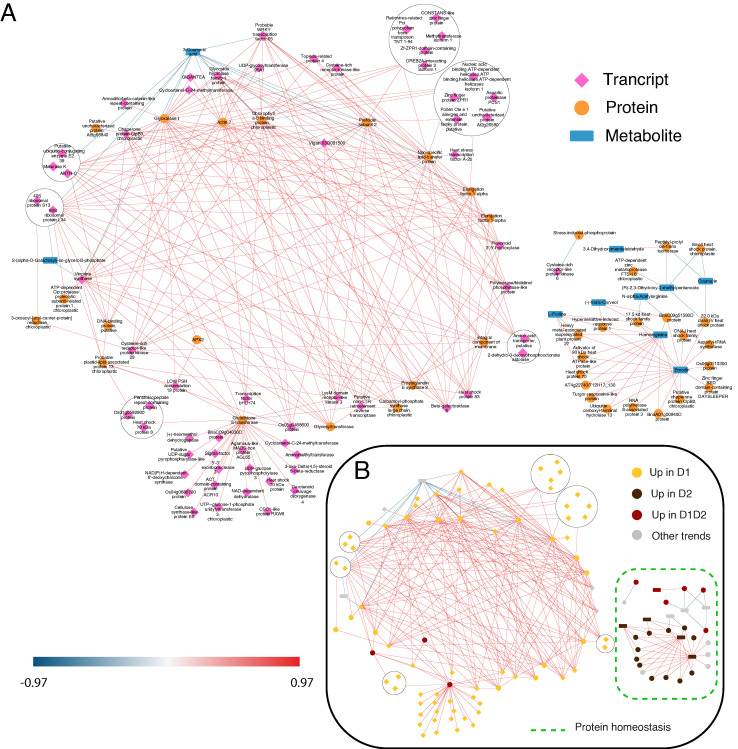
DIABLO network of the variables with more weight in the statistical analysis (threshold ± 0.9). (**A**) DIABLO network indicating the variables included in each omic layer. The three omic layers are represented in pink (transcripts), orange (proteins), and blue (metabolites). (**B**) DIABLO network indicating the variables included in the predominant trends. Positive connections are shown in red and negative connections in blue. D1 corresponds to 17 days of drought (early response, when chlorophyll fluorescence decreases by 20%), and D2 corresponds to 24 days of drought (later response, when chlorophyll fluorescence decreases by 45%). Predominant processes are rounded off with a dashed line.

**Figure 5 plants-11-03067-f005:**
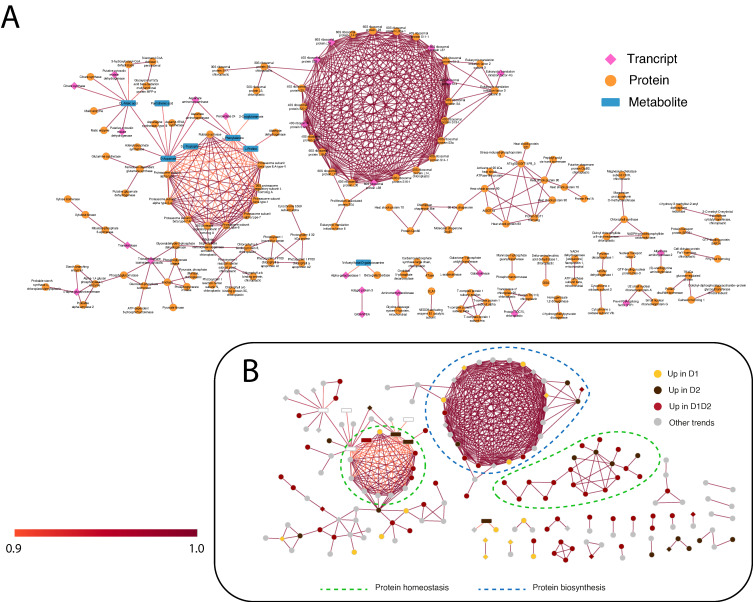
STITCH network of the variables with more weight in the functional analysis (confidence score > 0.9) using the *Arabidopsis thaliana* database. (**A**) STITCH network indicating the variables included in each omic layer. The three omic layers are represented in pink (transcripts), orange (proteins), and blue (metabolites). (**B**) STITCH network indicating the variables included in the predominant trends. The links between variables presented are the highest confidence (score > 0.9). The score is computed by combining the probabilities from the different evidence channels and corrected for the probability of randomly observing an interaction. D1 corresponds to 17 days of drought (early response, when chlorophyll fluorescence decreases by 20%), and D2 corresponds to 24 days of drought (later response, when chlorophyll fluorescence decreases by 45%). Predominant processes are rounded off with a dashed line.

## Data Availability

All datasets are available in the *Quercus ilex* web of our group (https://www.uco.es/probiveag/holm-oak-database.html) and in the supplementary material of this article. The transcriptome raw data are available in the NCBI’s Gene Expression Omnibus repository (GEO, accession number GSE145009). The protein raw mass spectrometry was deposited in the ProteomeXchange Consortium via the PRIDE partner repository [72] with the dataset identifier PXD017493. Metabolite raw data were deposited in the NIH Common Fund’s National Metabolomics Data Repository (NMDR) website, the MetabolomicsWorkbench, https://www.metabolomicsworkbench.org, where it has been assigned the project ID PR001361. The data can be accessed directly via the project DOI 10.21228/M8SQ6J.

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
