# Peer review of "Multiomic Data Integration in the Analysis of Drought-Responsive Mechanisms in Quercus ilex Seedlings"

_plants, 2022, doi:10.3390/plants11223067_

Round 1

Reviewer 1 Report

Holm oak (Quercus ilex L.) is the dominant tree species in the Mediterranean forest. The authors obtained -omics datasets (transcriptomics, proteomics, and metabolomics) from Q. ilex seedlings subjected to drought conditions under the typical summer conditions in Southern Spain. PCA and DIABLO were used for integrative analysis. Undoubtedly, the author provides valuable -omic data for further study on Q. ilex. However, the author needs to give the phenotype data of the stress treatment and validate the results through experiments such as qRT-PCR, etc.

1. The authors showed the -omic datasets in the manuscript. However, I cannot find the data you provided for reanalysis. Please ensure you upload your data to the public database, such as NCBI, ProteomeXchange and National Metabolomics Data Repository (NMDR).

2. In Figure 2, the author analyzed the -omic data by PCA first; however, the PCA results showed inconsistency among your replicates. It is not acceptable, and please recheck your data.

3. In Figure 3, the differential expression genes between D1 and D2 showed an entirely different pattern. Please compare these data carefully and explain this phenomenon.

4. Please provide the phenotype data for all your samples.

5. Please validate those genes involved in drought stress you proposed.

Author Response

Firstly, we really appreciate all the helpful comments and suggestions of the two reviewers. These comments have helped to improve the manuscript. Below we detail the changes and respond to each reviewer's questions.

Reviewer 1

Holm oak (Quercus ilex L.) is the dominant tree species in the Mediterranean forest. The authors obtained -omics datasets (transcriptomics, proteomics, and metabolomics) from Q. ilex seedlings subjected to drought conditions under the typical summer conditions in Southern Spain. PCA and DIABLO were used for integrative analysis. Undoubtedly, the author provides valuable -omic data for further study on Q. ilex. However, the author needs to give the phenotype data of the stress treatment and validate the results through experiments such as qRT-PCR, etc.

  1. The authors showed the -omic datasets in the manuscript. However, I cannot find the data you provided for reanalysis. Please ensure you upload your data to the public database, such as NCBI, ProteomeXchange and National Metabolomics Data Repository (NMDR).

# In the section "Data Availability Statement" are all the references and IDs from where the data used in this manuscript are uploaded.

“Data Availability Statement: All datasets are available in the Quercus ilex web of our group (https://www.uco.es/probiveag/holm-oak-database.html) and in supplementary material of this article. The transcriptome raw data is available in the NCBI’s Gene Expression Omnibus repository (GEO, accession number GSE145009). The Protein raw mass spectrometry were deposited on the ProteomeXchange Consortium via the PRIDE partner repository [72] with the dataset identifier PXD017493. Metabolite raw data were deposited in the NIH Common Fund’s National Metabolomics Data Repository (NMDR) website, the MetabolomicsWorkbench, https://www.metabolomicsworkbench.org, where it has been assigned Project ID PR001361. The data can be accessed directly via it's Project DOI: 10.21228/M8SQ6J.”

In addition, we have incorporated this information in each corresponding section in “Material and methods”.

  1. In Figure 2, the author analyzed the -omic data by PCA first; however, the PCA results showed inconsistency among your replicates. It is not acceptable, and please recheck your data.

# Holm oak is characterized by high phenotypic variability, either inter and intrapopulation, which correlates which differences in gene expression and biomolecule profiles. Also, and because is allogamous, there are differences between brother seedlings. This is reflected in our past publications based on transcriptomics, proteomics, and metabolomics analysis. We have used seedlings from seeds of a mother tree from of “Almadén de la Plata” (Seville, Andalusia, Spain; 37° 52′ N, 6° 28′ W). The total and original dataset resulted in a PCA (Fig. 1A) with replicates dispersion for some samples, but grouping and discrimating the samples.  We knew, and because of that we are very conservative in our conclusions, based in consistent differences clearly associated to the treatment and time point. For that, we reduced the variability by eliminating the noise coming from data with a high coefficient of variation, the PCA made with the variables significant at p<0.05. As you can see in Fig. 1B, it resulted in lower variability and good grouping of replicates.

We have reviewed our data, and have verified that the clustering is correct, seeing that even at lower levels of p<value a similar clustering (see attached image in the PDF (response_reviewer1.pdf).

  1. In Figure 3, the differential expression genes between D1 and D2 showed an entirely different pattern. Please compare these data carefully and explain this phenomenon.

# In Figure 3 we see the three patterns, not only the differentials between D1 and D2, there is also a very abundant pattern corresponding to overexpression/ accumulation in both sampling times (D1&D2). The existence of differential patterns in the response to drought stress is something that has already been seen in previous studies [1–3], and it is not surprising considering the existence of early (D1) and late (D2) responses, it depending of the level of stress at that moment. So we can conclude on transient (observed at just one time) or common (observed at both times) responses, gene expression and biomolecule profile. In addition it is well known the existence of the time course corresponding to the mRNA, protein, and metabolite peaks.

  1. Please provide the phenotype data for all your samples.

# Six-month-old seedlings of the drought tolerant species Quercus ilex were used. Open pollinated seedlings were between 10-15 cm tall with around 8-10 adult leaves.

Adult leaves without stress symptoms were analyzed at two sampling points: when leaf florescence decreased 20 % (D1, after 17 days in drought) and 45% (D2, after 24 day in drought). We have included in the manuscript all the phenotypic data of the plants used for sampling. In any case, and as reported in San-Eufrasio et al. [4] Q. ilex is one of the most drought tolerant Quercus species. Even so, there are differences in the level of tolerance among populations and individuals. The seedlings used for the molecular analysis did not show damage symptoms even though the severe drought imposed, so these individuals can be catalogued from a phenotypic point of view as highly tolerant.

  1. Please validate those genes involved in drought stress you proposed.

# Unfortunately, there is no material left to perform these analyses. Given the phenology of the species, repeating the experiment (from seedling collection, assay and analysis) would take more than a year.

  1. San-Eufrasio, B.; Castillejo, M.Á.; Labella-Ortega, M.; Ruiz-Gómez, F.J.; Navarro-Cerrillo, R.M.; Tienda-Parrilla, M.; Jorrín-Novo, J. V; Rey, M.-D. Effect and Response of Quercus ilex subsp. ballota [Desf.] Samp. Seedlings From Three Contrasting Andalusian Populations to Individual and Combined Phytophthora cinnamomi and Drought Stresses. Front. Plant Sci. 2021, 12, 722802.
  2. Tienda-Parrilla, M.; López-Hidalgo, C.; Guerrero-Sanchez, V.M.; Infantes-González, Á.; Valderrama-Fernández, R.; Castillejo, M.-Á.; Jorrín-Novo, J. V; Rey, M.-D. Untargeted MS-Based Metabolomics Analysis of the Responses to Drought Stress in Quercus ilex L. Leaf Seedlings and the Identification of Putative Compounds Related to Tolerance. Forests 2022, 13.
  3. Guerrero-Sánchez, V.M.; Castillejo, M.Á.; López-Hidalgo, C.; Alconada, A.M.M.; Jorrín-Novo, J.V.; Rey, M.-D. Changes in the transcript and protein profiles of Quercus ilex seedlings in response to drought stress. J. Proteomics 2021, 243, 104263.
  4. San-Eufrasio, B.; Sánchez-Lucas, R.; López-Hidalgo, C.; Guerrero-Sánchez, V.M.; Castillejo, M..; Maldonado-Alconada, A.M.; Jorrín-Novo, J.V.; Rey, M.-D. Responses and differences in tolerance to water shortage under climatic dryness conditions in seedlings from Quercus spp. and Andalusian Q. ilex populations. Forests 2020, 11, 707.

Reviewer 2 Report

  1. The authors did a good job of understanding the drought-responsive mechanisms in Quercus ilex seedlings by integrating proteomics, transcriptomics, and metabolomics.
  2. Start the abstract with a clear statement on the scope, relevance, and intention of the study before describing the main results. End the abstract with a clear statement about the main conclusions and perspectives of the work.
  3. Many sentences do not make sense and should be clearly rewritten. Authors are urged to go through the text and correct it. The quality of the english language and grammar for the entire manuscript needs to improve by native English speakers.
  4. If possible, restructure and carefully edit the results and discussion sections and add clear information regarding the most noteworthy findings.  The author should mainly discuss more how similar/dissimilar this study's result is with previous studies.
  5. I suggest validating the genes by qPCR  analysis

Author Response

Firstly, we really appreciate all the helpful comments and suggestions of the two reviewers. These comments have helped to improve the manuscript. Below we detail the changes and respond to each reviewer's questions.

1. The authors did a good job of understanding the drought-responsive mechanisms in Quercus ilex seedlings by integrating proteomics, transcriptomics, and metabolomics.

2. Start the abstract with a clear statement on the scope, relevance, and intention of the study before describing the main results. End the abstract with a clear statement about the main conclusions and perspectives of the work.

# The abstract has been modified according to the reviewer's suggestions.

3. Many sentences do not make sense and should be clearly rewritten. Authors are urged to go through the text and correct it. The quality of the english language and grammar for the entire manuscript needs to improve by native English speakers.

# English has been revised and modified through mdpi's language editing service.

3. If possible, restructure and carefully edit the results and discussion sections and add clear information regarding the most noteworthy findings.  The author should mainly discuss more how similar/dissimilar this study's result is with previous studies.

# It has rewritten including phrases emphasizing what has been seen in previous studies (such as the role of proteases and chaperones) and what is new in this article, such as the transcriptomic response in the early response to drought stress.

4. I suggest validating the genes by qPCR  analysis

# Unfortunately, there is no material left to perform these analyses. Given the phenology of the species, repeating the experiment (from seedling collection, assay and analysis) would take more than a year.

Round 2

Reviewer 1 Report

I believed that Quercus ilex's high phenotypic variability could be acceptable.
To confirm the variations, qRT-PCR and phenotypic data (pictures) are still required.

Reviewer 2 Report

Accept in current form. Language needs to be polished.